# A Simple and Robust Spectral Index for Identifying Lodged Maize Using Gaofen1 Satellite Data

**DOI:** 10.3390/s22030989

**Published:** 2022-01-27

**Authors:** Yuanyuan Chen, Li Sun, Zhiyuan Pei, Juanying Sun, He Li, Weijie Jiao, Jiong You

**Affiliations:** 1Academy of Agricultural Planning and Engineering, Ministry of Agriculture and Rural Affairs, Beijing 100121, China; chenyuanyuan@aape.org.cn (Y.C.); peizhiyuan@aape.org.cn (Z.P.); sunjuanying@aape.org.cn (J.S.); jiaoweijie@aape.org.cn (W.J.); youjiong@aape.org.cn (J.Y.); 2Big Data Development Center, Ministry of Agriculture and Rural Affairs, Beijing 100125, China; 3State Key Laboratory of Resources and Environmental Information System, Institute of Geographic Science and Natural Resources Research, Chinese Academy of Sciences, Beijing 100101, China; lih@Ireis.ac.cn

**Keywords:** maize lodging, spectral feature, Gaofen1, index

## Abstract

Crop lodging is a major destructive factor for agricultural production. Developing a cost-efficient and accurate method to assess crop lodging is crucial for informing crop management decisions and reducing lodging losses. Satellite remote sensing can provide continuous data on a large scale; however, its utility in detecting lodging crops is limited due to the complexity of lodging events and the unavailability of high spatial and temporal resolution data. Gaofen1 satellite was launched in 2013. The short revisit cycle and wide orbit coverage of the Gaofen1 satellite make it suitable for lodging identification. However, few studies have explored lodging detection using Gaofen1 data, and the operational application of existing approaches over large spatial extents seems to be unrealistic. In this paper, we discuss the identification method of lodged maize and explore the potential of using Gaofen1 data. An analysis of the spectral features after maize lodging revealed that reflectance increased significantly in all bands, compared to non-lodged maize. A spectral sum index was proposed to distinguish lodged and non-lodged maize. Two study areas were considered: Zhaodong City in Heilongjiang Province and Ningjiang District in Jilin Province. The results of the identified lodged maize from the Gaofen1 data were validated based on three methods: first, ground sample points exhibited the overall accuracies of 92.86% and 88.24% for Zhaodong City and Ningjiang District, respectively; second, the cross-comparison differences of 1.01% for Zhaodong City and 1.13% for Ningjiang District were obtained, compared to the results acquired from the finer-resolution Planet data; and third, the identified results from Gaofen1 data and those from farmer survey questionnaires were found to be consistent. The validation results indicate that the proposed index is promising, and the Gaofen1 data have the potential for rapid lodging monitoring.

## 1. Introduction

Crop lodging, which is the bending of plant stems from their upright position or the destruction of the plant root–soil anchorage system, is usually caused by strong storms or heavy rains induced by typhoons [1,2]. Crop lodging, which occurs after the passing of a strong typhoon, has been recognized as a major agricultural disaster. Lodging is detrimental to biomass accumulation; therefore, crop yield usually decreases and grain quality declines after lodging occurs [3,4,5]. Since they are relatively tall, cereal crops, especially maize, are more easily affected by lodging [6,7]. According to the trials conducted over 16 locations and 11 years in Texas, Farfan et al. revealed that lodging was negatively correlated (approximately 15% to 21%) with maize yield [8]. Various studies reported that maize yield losses could range from 15–30% or even reach 50% in severe lodging regions [9,10]. In addition, lodging is a major factor affecting crop harvesting [11]. The economic losses induced by lodging are enormous. Therefore, timely, rapid, and quantitative identification of maize lodging areas is important and is the basis of loss assessments.

The traditional strategies for crop lodging assessments are generally field-/lab-based, and these strategies are often unfeasible due to the constraints associated with limited coverage and labor intensive nature of the work. As a result of its large spatial coverage, satellite remote sensing (RS) is playing an increasingly important role in crop monitoring and offers a more scalable approach. The principle of RS-based crop monitoring is that the reflectance or backscatter coefficient at different wavelengths is affected by changes in the crop canopy. As a result of the complexity of crop lodging, the number of papers identifying crop lodging using satellite RS is limited. Chauhan et al. documented the retrieval of RS-based crop lodging assessment information [3]. According to a review paper by Chauhan et al., research in this field has been going on since 2015, with Yang et al.’s study being regarded as the first article to demonstrate the potential of using satellite radar data to detect lodged wheat [3,12]. To the best of our knowledge, only approximately 20 articles researching crop lodging monitoring-based satellite RS technology have been published since 2015. These studies explore the use of sensors onboard Sentinel-1 and -2, Radarsat-2, HJ_1, and Worldview-2 satellites [2,13,14,15,16,17,18]. The study objects mainly include wheat, maize, rice, and sugarcane [7,12,13,19,20,21,22]. The limited number of studies focusing on crop lodging detection using satellite RS demonstrates that this field is still in the experimental stage. Fortunately, there has been an increasing interest in recent years [3], and this field is evolving.

In review of the aforementioned studies, vegetation indexes (VIs), such as the normalized difference vegetation index (NDVI), the ratio vegetation index (RVI), etc., were commonly adopted to discriminate lodged crops [9,21]. Moreover, a clear upward tendency of reflectance in the multi-spectral bands of lodged crops is noted [16,17]. If the overall reflectance increases after crop lodging, the variation of these VIs would not be obvious, and thus VIs may not be a good method with which to distinguish the lodged and non-lodged maize. Therefore, the crop lodging spectral behavior and detection methods need to be further analyzed.

China launched the High-Resolution Earth Observation System in 2010. Gaofen1 (GF-1), which was launched in 2013, is the first satellite in the Gaofen series. One of GF-1′s main contributions is to perform agricultural monitoring [23,24]. The free accessibility, high temporal-spatial resolution and wide coverage of GF-1 data make it possible to develop operational applications for crop lodging identification. However, few articles focus on the spectral changes between lodged and non-lodged maize using GF-1 data. We found only one published paper identifying lodged maize using GF-1 images [25]. In the paper, the lodging grade evaluation index was constructed using the measured lodging proportion and angle. However, the evaluation index cannot be directly calculated using the remote sensing data and was retrieved indirectly using an optimal combination of variables, including VIs and reflectance in various bands. This implies that this method was also based on VIs, as mentioned above. Another relevant problem is that the optimal combination of variables may not be suitable when applied to other areas. These issues beg the questions: are VIs effective for monitoring maize lodging? Additionally, is there a more suitable method?

This paper aimed to: (i) explore how lodging incidence affects the reflectance of maize; (ii) propose a simple identification method that is not NDVI-based to distinguish lodged and non-lodged maize; and (iii) assess whether the proposed method is practical and operational when applied to different areas.

## 2. Study Area and Data

### 2.1. Study Area

Northeast China is a substantial food production base, and is often affected by natural disasters. The strong typhoon “Meishake”, which has a maximum wind power grade above 10, caused a landfall in Northeast China on 3–4 September 2020. Large areas of maize, which were in the milking and maturation stages, lodged, especially in a 100 km area in the typhoon path in Heilongjiang Province and Jilin Province. However, acquiring high spatial resolution imagery covering the entire aforementioned region in a short period is not realistic. Therefore, two study areas (shown in Figure 1), Zhaodong City in Heilongjiang Province and Ningjiang District in Jilin Province, were selected to assess the application of GF-1 data in monitoring the lodged maize. Both areas are known for maize production and are near the typhoon “Meishake” path. The maize planting area accounts for nearly 60% of the farmland in Zhaodong City and 42% in Ningjiang District. Zhaodong City, covering an area of 4332 km^2^ in the region from 125°22′–126°22′ E and 45°10′–46°25′ N, is characterized by a cold temperate climate with a long and cold winter. Ningjiang District, located from 124°36′–125°15′ E and 44°55′–45°32′ N, has an area of 1313 km^2^ with a temperate continental monsoon climate, i.e., hot and rainy summers and cold and dry winters. Songhua River passes through the Ningjiang District.

### 2.2. GF-1 Satellite Data

The wide-field-view (WFV) sensor, which is a payload onboard the GF-1 satellite, has a 16 m spatial resolution and 4 channels covering the blue, green, red and near-infrared bands. The GF-1 satellite has a short revisit cycle of 4 days and a wide scanning swath of 800 km, which is important for observing the lodging phenomenon since lodging usually occurs instantly and over a large area. Table 1 provides an overview of the relevant GF-1 data [26]. Two images for Zhaodong City and one for Ningjiang District retrieved on 6 September 2020 were used to study the spectral characteristics of lodged maize and distinguish lodged maize from non-lodged maize in this paper. The rapid mapping of lodged maize requires knowledge of the spatial distribution of normal maize (before lodging). To dismiss the effect of lodging on extracting the spatial distribution of normal maize, images retrieved on 16 and 20 August 2020 were used to map the spatial distribution of normal maize. All GF-1 data were processed with a radiometric correction, atmospheric correction, and geometric correction using the Remote Sensing Desktop and Envi software.

### 2.3. Planet Satellite Data

To evaluate the performance of GF-1 data application in identifying lodged maize, the top of atmosphere reflectance product from Planet Dove satellites, which possess a finer resolution than the GF-1 satellite at 3 m and similar bands, as shown in Table 1, was obtained from a commercial company. Planet Dove is a low-orbital (475 km altitude) constellation composed of more than 150 operating satellites [27]. Planet satellites acquire images in narrow swaths of 24.6 km × 16.4 km [28]. Given the time of the lodging event observed in this study, Planet data were gathered on 5, 6, and 11 September 2020. The mosaic image in Zhaodong City was produced using 42 non-cloudy scenes. Only one good-quality scene was obtained in Ningjiang District. As a result of the narrow swaths of the Planet satellites, the study areas were not totally covered by Planet data. In other words, 83% of the land area in Zhaodong City and 20% in Ningjiang District were covered by Planet data. Figure 2 shows the coverage scope of Planet data.

### 2.4. Ground True Information

The use of satellite RS to monitor crop conditions without ground truth information is not reliable. Using the in situ observation, we found that the maize inclination angle in the vertical direction was almost greater than 60° in most lodging areas; in other words, the maize nearly completely lodged. Thus, this paper focused on the identification of lodged and non-lodged maize. Further lodging severity according to inclination angle was not explored. Two ground data sources collected from 6–9 September 2020 were used as the ground truth information. The first involved investigating in the field for the lodging or non-lodging attribute of maize. Two hundred and sixty-three sample points (140 for Zhaodong and 123 for Ningjiang) in the completely lodged areas and non-lodged areas were generated. A portable sub-meter GPS was used to position each sample point and the lodging or non-lodging attribute for each sample point was recorded. At these points, approximately 70% of the data (98 in Zhaodong City and 89 in Ningjiang District, denoted the training samples) was used for studying the spectral response of the lodged maize; the remaining data (denoted the validating samples) were used for validating the results of lodged maize identification. To clearly show the image tone features of lodged and non-lodged maize, Figure 3 displays the distribution of the partial sample points. The points located in lodged areas were considered lodged points (shown in yellow), and those located in non-lodged areas were considered non-lodged points (shown in blue).

In terms of the second ground data source, a survey of farmers focusing on the ratio of lodged maize area to planted maize area in the family unit (labeled as the lodging proportion) was conducted. Five lodging proportion grades: 0–20%, 21–40%, 41–60%, 61–80%, and 81–100%, were set. Each respondent was instructed to choose one grade to represent the lodging maize proportion in his/her family unit. These data from survey questionnaires were used for the purpose of result validation.

## 3. Method

The workflow for identifying lodged maize is plotted in Figure 4. First, the area and spatial distribution of normal maize were extracted using the support vector machine (SVM) method, which is popular in the image classification field. Second, the spectral features of lodged maize were analyzed with the help of in situ sample points, and a spectral sum index was established to detect lodged maize with an appropriate threshold. The dichotomy map of the lodged and non-lodged region was obtained using the established index. The map of lodged maize was produced by masking out the non-maize region using the map of normal maize. Finally, the results of the lodged maize areas were assessed using in situ sample points, Planet data, and field surveys from farmers.

### 3.1. Maize Mapping

Several studies were conducted on maize extraction using optical satellite data, and relatively mature extraction methods were developed [29,30]. The SVM method, a non-parametric algorithm, is widely used to perform maize mapping with a high accuracy [26,31,32,33]. Considering the crop calendar and avoiding the lodging effect, the images taken before the lodging event were used to map the spatial distribution of maize using the SVM classifier. The regions of interest (ROIs) were selected based on a priori knowledge of the multi-year agricultural remote sensing monitoring business. Five hundred validation points for each study area were randomly produced in the ArcGIS software to evaluate the maize extraction results. The overall accuracies of 92.6% in Zhaodong City and 94.8% in Ningjiang District were achieved, which can meet the requirement of discriminating the lodged and non-lodged maize.

### 3.2. Theoretical Analysis of Lodged Maize Reflectance

We can see in Figure 3 that the regions in which yellow points are located appear in a pink tone in the RGB image, while the areas in which blue points are located appear in a dark red tone. The tone change, which resulted from reflectance variations, is a critical manifestation of lodged maize in RGB images. The change in the lodged maize reflectance may be explained in three ways: first, the canopy structure was changed when the lodging incidence occurred. Specifically, a normal maize canopy is characterized by stems and leaves standing upward, while lodged maize canopy is composed of stacked stems and leaves, which manifests as a flatted surface [34]. A flattened and denser canopy enlarges the reflective surface and decreases the roughness of the canopy surface, leading to an increase in reflectance [35]. Second, with the increasing canopy coverage in a lodging event, the contribution of soil and shaded leaves, which have a lower reflectance, is decreased [36]. Third, the change in the biophysical parameters in the lodged maize may cause the reference change, e.g., the slow declination of the chlorophyll content and leaf water content as photosynthesis is disrupted would reflect more energy than normal maize [16,37,38,39]. The above analyses suggest that the reflectance from visible to near-infrared band of lodged maize may be obviously higher than that of normal maize. For the confirmation of the theoretical analyses, the reflectance of lodged and non-lodged maize, which was defined by the training samples described in Section 2.4, was extracted from the GF-1 images and then the box plot was drawn.

### 3.3. Lodged Maize Detection

According to the above theoretical analysis, the reflectance in all bands following maize lodging is similar to the findings of previous studies [17,21,25], in which VIs were used to identify lodged maize. We know that VIs can enhance vegetation change information when the reflectance in relative bands changes with different features. In this paper, the continuous increase in spectral reflectance for the overall bands indicated that applying the spectral sum could further strengthen the difference between lodged and non-lodged maize. Thus, a lodged maize spectral sum index (SSI) was constructed by summing the reflectance of the four bands as follows:(1)$$SSI=R_{1}+R_{2}+R_{3}+R_{4}$$
where *R*_1_, *R*_2_, *R*_3_, and *R*_4_ are the reflectance of the blue, green, red, and near-infrared bands, respectively.

Since the reflectance increases for all bands, the appropriate threshold of SSI, which is needed to distinguish lodged and non-lodged maize, can be easily determined. As shown in Figure 5, there was a boundary (marked by the blue dotted line) during the values of SSI presented by the lodged and non-lodged maize. Thus, the threshold value of 0.6200 was set for Zhaodong City and Ningjiang District in this study. The pixels whose reflectance sums were higher than the threshold were distinguished as a lodged region; otherwise, they were distinguished as a non-lodged region. Lodged maize maps were produced by masking the lodged region using the map of the maize spatial distribution generated in Section 3.1.

### 3.4. Validation of Results

The accuracies of the determined lodged maize regions were validated using three methods: first, they were validated based on the validating samples. An error matrix, which is the classic method to assess the mapping accuracy of remote sensing images, was constructed to evaluate the lodged maize accuracy, with the properties of validating samples as references. Forty-two validation points in Zhaodong City and thirty-four in Ningjiang District were used to generate the error matrix. The user’s accuracy (UA), producer’s accuracy (PU), and overall accuracy calculated based on the error matrix were adopted to evaluate the identification results.

Second, the lodged maize areas acquired from GF-1 images were cross-compared to those obtained from the finer-resolution Planet data. Since the same change features that the overall reflectance increased were also obtained from the Planet data after maize lodging, the SSI method was applied to the Planet data with a different appropriate threshold. The cross-comparison difference of the identified lodged maize was calculated using Equation (2) as follows:(2)$$D=\frac{S_{GF}-S_{p}}{S_{p}}*100\%$$
where *D* represents the cross-comparison difference of the lodged maize region results and is labeled *D*_zhaodong_ for Zhaodong City and *D*_ningjiang_ for Ningjiang District; *S_GF_* and *S_P_* are the areas of lodged maize extracted from the GF-1 and Planet data, respectively.

Third, the lodging proportions derived from satellite data were verified using the farmer survey results. The lodging proportion grade, which was selected by most respondents, was used as the reference information to evaluate whether the lodging proportions derived from GF-1 data are within this grade or not.

## 4. Results

### 4.1. Reflectance of Lodged Maize

Figure 6 clearly shows the spectral response of the lodged maize. We can see that the box of lodged maize (showing with slashed background) is higher than that of non-lodged maize (showing without slashed background) in each band. This means that most reflectance from the blue to near-infrared band representing the lodged maize is higher than that of non-lodged maize. This result verifies the analysis for the reflectance change in lodged maize, compared with non-lodged maize in Section 3.2. For the specific results, the mean values of the spectral reflectance representing lodged and non-lodged maize in each band were also shown in Figure 6. Figure 6 reveals that the lodged and non-lodged plots are analogous, but there is a clear upward trend in the lodged plots. The reflectance of lodged maize substantially increased in all bands as compared to non-lodged maize. The spectral changes were similar in both study areas but had different values. As regards Zhaodong City, the mean reflectance increased from 0.0188 to 0.0400 in the blue band, 0.0496 to 0.0829 in the green band, 0.0327 to 0.0639 in the red band, and 0.4703 to 0.5389 in the near-infrared band. For Ningjiang District, the same changes were 0.0241 to 0.0346, 0.0613 to 0.0803, 0.0452 to 0.0613, and 0.4346 to 0.5435 in the four bands, respectively. Table 2 provides the reflectance increments and amplifications. The maximum increment appeared in the near-infrared band, followed by the green band for both study areas. In terms of amplification, the values in the blue and red bands were more pronounced.

### 4.2. Lodged Maize Area

Figure 7 displays the spatial distributions of lodged and non-lodged maize in the two study areas. The lodged maize area of Zhaodong City was 112.17 thousand hectares (ha), which accounted for 54.56% of the entire maize planting area of 205.59 thousand ha. In the case of Ningjiang District, the lodged maize area and maize planting area were 41.72 and 51.01 thousand ha, respectively, and the lodged proportion was 81.79%. We can see that the lodged area exceeded 50% for both study areas, especially in Ningjiang District, for which the percentage was as high as 80%. It can be concluded that lodging was severe, with a large lodged area observed in the typhoon-affected region.

### 4.3. Validation

#### 4.3.1. Validating Samples

By analyzing the property of each validating sample recorded in the field and the identified results using GF-1 data, the error matrix was generated, as shown in Table 3. With regard to Zhaodong City, the UA and PA of the lodged maize and non-lodged maize were both more than 90%. Additionally in Ningjiang District, the UA and PA for the lodged maize were both 90%, and they were 85.71% for the non-lodged maize. The overall accuracies were 92.86% and 88.24% for Zhaodong City and Ningjiang District, respectively.

#### 4.3.2. Cross-Comparison

In terms of the cross-comparison with the Planet data, we obtained the following results: (i) for Zhaodong City, the lodged maize area obtained from the Planet data was 88.97 thousand ha, contrasting with the area of 89.87 thousand ha obtained from the GF-1 data in the same region. As calculated by Equation (2), D_zhaodong_ was 1.01%; (ii) for Ningjiang District, a lodged maize area of 8.87 thousand ha was obtained from the Planet data, contrasting with the area of 8.97 thousand ha obtained from the GF-1 data in the same region, producing a D_ningjiang_ value of 1.13%. The lodged maize identification results obtained using the GF-1 data were consistent with the results obtained from the Planet data, indicating that the SSI method was feasible for identifying lodged maize. With its wide swath, short revisit period, and free accessibility, GF-1 data can serve as the main satellite data to detect maize lodging events, and Planet data can serve as auxiliary information.

#### 4.3.3. Farmer Surveys

The results of the surveys given to farmers are shown in Table 4. We collected 18 questionnaires from farmers in Zhaodong City and 17 in Ningjiang District. For Zhaodong City, a lodging proportion grade of 41–60% was selected in 7 questionnaires, accounting for nearly 40% of the 18 questionnaires. Each of the other lodging proportion grades was chosen in less questionnaires. In Ningjiang District, more than half of the questionnaires, 9 out of 17, selected a lodging proportion grade of 81–100%. The survey results stated that the lodged maize area obtained based on the GF-1 data was credible and that the identification method was practicable.

### 4.4. Comparison of SSI with NDVI

To further study the VIs change feature between the lodged and non-lodged maize, the NDVI was calculated using the red and near-infrared bands extracted from the pixels used to calculate *SSI* and plotted in Figure 8. We can see that there is no clear dividing line to discriminate the values of NDVI for lodged and non-lodged maize. This is because, for most of the pixels corresponding to the lodged maize, both references in red and near-infrared bands increased, while the minus sign in the calculation formula of NDVI could weaken the spectral change trend of the simultaneously increasing trend in red and near-infrared bands. Thus, the change in NDVI was not obvious. After maize lodging, the growth was inhibited to a certain extent, which led to the weak downward trend of NDVI. The mean NDVI of lodged maize was lower than that of non-lodged maize, as shown in Table 5. This conclusion was the same as in previous studies [17,21,34]. Despite this, the change in NDVI was far lower than the change in *SSI*.

## 5. Discussion

This paper aimed to study the effect of maize lodging on the spectral response and to propose a practical and straightforward method for detecting lodged maize. The critical change in maize after lodging was that the reflectance curve had a clear upward trend. Different from previous studies, in which the variation of VIs was the basis of lodged maize detection [9,21,25], we summed up the reflectance and proposed *SSI* method to further enhance the spectral difference between lodged and non-lodged maize. To our knowledge, there is no analogous method in the literature to detect lodged maize. The manifest difference between lodged and non-lodged maize in the *SSI* image could be clearly observed. Thus, it would not be challenging to set an optimal threshold to discriminate between lodged and non-lodged maize. If there is no available in situ data, considering the obvious tone change of GF-1 image between lodged and non-lodged maize, the ROIs of lodged and non-lodged maize can be chosen on the images retrieved after lodging event. According to *SSI* values calculated using the pixel reflectance within the ROIs, the box plot can be drawn. Then, the appropriate threshold to discriminate lodged and non-lodged maize will be obtained based on the box plot. Alternatively, the threshold can be calculated by maximizing the variance of *SSI* values within the ROIs of lodged and non-lodged maize.

The main advantages of the *SSI* method are as follows: (i) it is more efficient and inexpensive than the field survey method; (ii) it produces the lodged maize map quickly in a simple operation; and (iii) it can be used in other areas to identify lodged maize at the milking and maturation stages without the help of in situ data. Despite the promising results regarding the detection of lodged maize, the acquisition of optical GF-1 imagery at large spatial extents that coincide with specific times may not always be feasible, since optical data are often limited by clouds and rain. This problem may be solved by merging optical and microwave data [40]. It is important to note that research into crop lodging detection and assessment using satellite RS techniques is still in its infancy [41], and thus substantial research effort is still required.

## 6. Conclusions

This study interpreted the spectral changes resulting from maize lodging and proposed a method for quickly mapping lodged maize. The potential application of GF-1 data in the identification of lodged maize was assessed. We found a significant increase in the entire spectrum for lodged maize as compared with non-lodged maize. The *SSI* was established in our study, and the appropriate threshold could be easily determined. To assess the robustness of the *SSI* method, the *SSI* index was applied to two study areas: Zhaodong City in Heilongjiang Province and Ningjiang District in Jilin Province. The areas of lodged maize were 112.17 and 41.72 thousand ha in Zhaodong City and Ningjiang District, respectively. The validation based on the validating samples presented the overall accuracies of 92.86% in Zhaodong City and 88.24% in Ningjiang District for the lodged and non-lodged maize detection. The cross-comparison of results for lodged maize suggested that GF-1 data are feasible for lodged maize detection, and Planet data can be used in a supplementary manner or for verification. The lodged maize proportions of 54.56% in Zhaodong City and 81.79% in Ningjiang District also demonstrated an agreement with the survey’s results. By comparing *SSI* with NDVI, we concluded that the change in *SSI* after maize lodging was more pronounced than that in NDVI. These pieces of evidence confirm that *SSI* method can be used to extract the lodged maize, and GF-1 data exhibit a good performance in lodged maize identification.

Future effort should be directed towards merging GF-1 and microwave data, and exploring the features of lodging events in the merged data [18]. Given that different lodging severities can lead to different yield losses, lodging severity derived by the inclination angle should receive more attention according to the different lodging events.

## Figures and Tables

**Figure 1 sensors-22-00989-f001:**
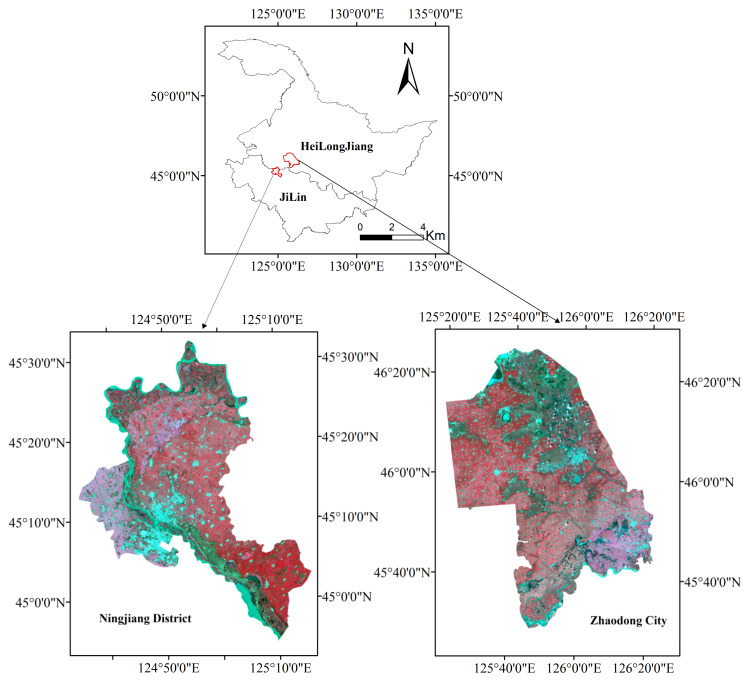
Study areas with false-color images (R: band 4; G: band 3; B: band 2).

**Figure 2 sensors-22-00989-f002:**
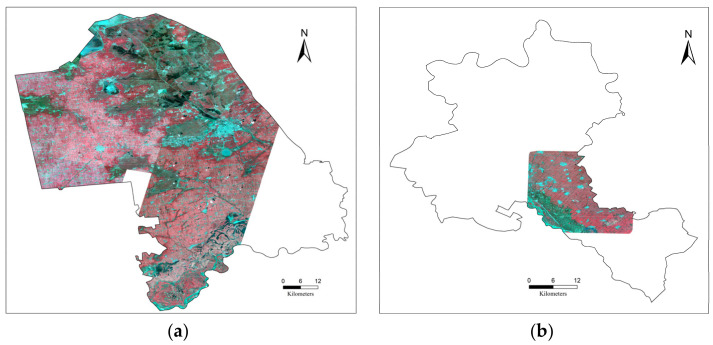
The scope of the Planet data coverage in a false-color image (R: band 4; G: band 3; and B: band 2) in (**a**) Zhaodong City and (**b**) Ningjiang District.

**Figure 3 sensors-22-00989-f003:**
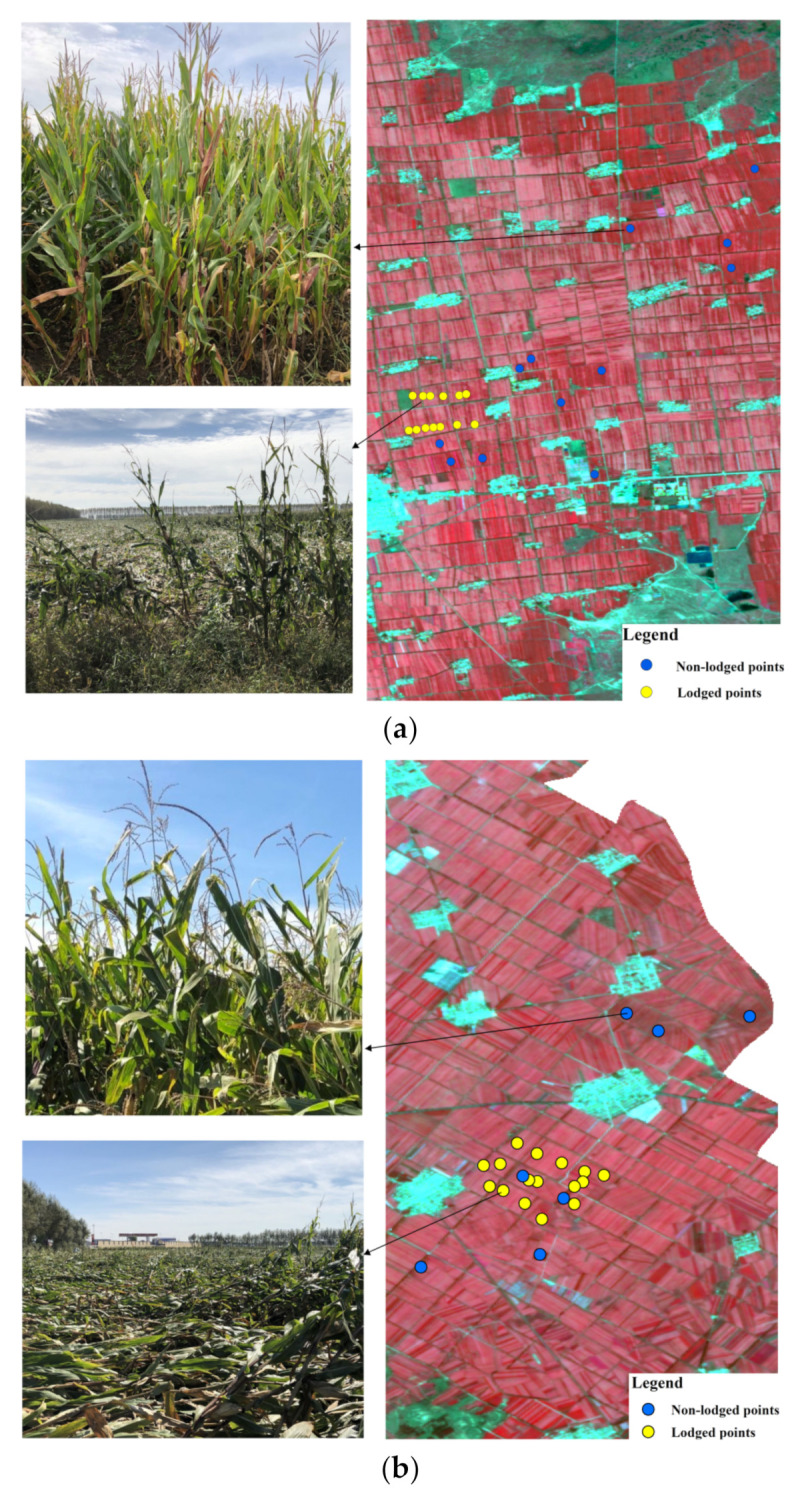
Distribution of sample points overlaid on Gaofen1 images in (**a**) Zhaodong City and (**b**) Ningjiang District.

**Figure 4 sensors-22-00989-f004:**
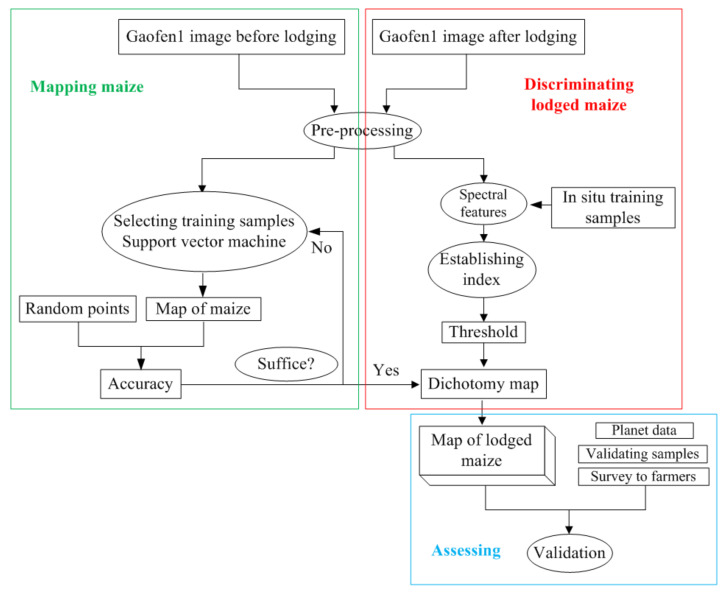
The workflow of lodged maize identification.

**Figure 5 sensors-22-00989-f005:**
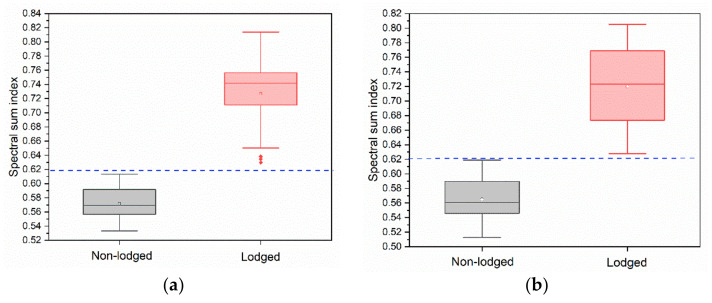
The value of spectral sum calculated according to the training samples for lodged and non-lodged maize in (**a**) Zhaodong City and (**b**) Ningjiang District. The value pointed by the blue dotted line represents the appropriate threshold to discriminate lodged and non-lodged maize.

**Figure 6 sensors-22-00989-f006:**
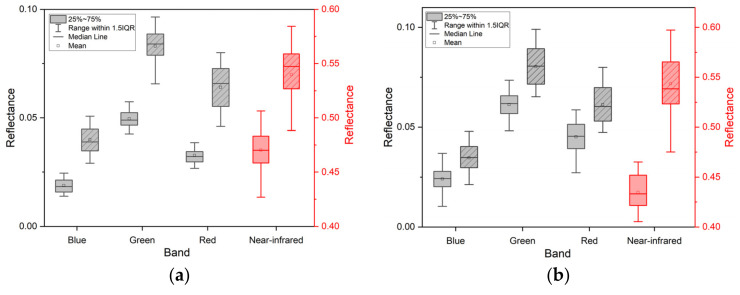
Box plots of spectral reflectance for lodged (shown with a slashed background) and non-lodged maize (shown without a slashed background) in (**a**) Zhaodong City and (**b**) Ningjiang District. The black color is for the blue, green, and red bands; the red color is for the near-infrared band.

**Figure 7 sensors-22-00989-f007:**
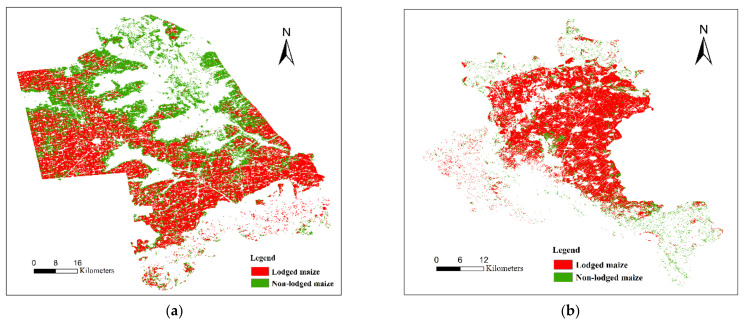
Distribution maps of the lodged and non-lodged maize in (**a**) Zhaodong City and (**b**) Ningjiang District.

**Figure 8 sensors-22-00989-f008:**
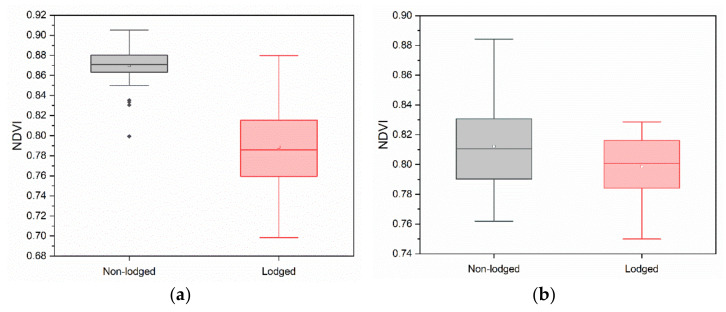
The values of the normalized difference vegetation index for the lodged and non-lodged maize in (**a**) Zhaodong City and (**b**) Ningjiang District.

**Table 1 sensors-22-00989-t001:** The parameters of the satellite data used in this paper.

Satellite Type	Band Number	Band	Wavelength Range (µm)	Spatial Resolution (m)
GF-1 WFV	1	Blue	0.45–0.52	16
	2	Green	0.52–0.59	
	3	Red	0.63–0.69	
	4	Near-infrared	0.77–0.89	
Planet	1	Blue	0.46–0.52	3
	2	Green	0.50–0.59	
	3	Red	0.59–0.67	
	4	Near-infrared	0.78–0.86	

**Table 2 sensors-22-00989-t002:** The spectral reflectance variation in lodged maize compared to the non-lodged maize.

Band	Zhaodong City		Ningjiang District	
	Increment	Amplification (%)	Increment	Amplification (%)
Blue	0.0212	112.77	0.0105	43.57
Green	0.0333	67.14	0.0190	31.00
Red	0.0312	95.41	0.0161	35.62
Near-infrared	0.0686	14.59	0.1089	25.06

**Table 3 sensors-22-00989-t003:** Error matrix between the identified results of the lodged and non-lodged maize and the properties of validating samples.

	Identified Results	Zhaodong City			Ningjiang District		
Properties of Validating Samples		Lodged Maize	Non-Lodged Maize	Producer’s Accuracy (PA, %)	Lodged Maize	Non-Lodged Maize	Producer’s Accuracy (PA, %)
Lodged maize		20	1	95.24	18	2	90
Non-lodged maize		2	19	90.48	2	12	85.71
User’s accuracy (UA, %)		90.91	95		90	85.71	
Overall accuracy		92.86%			88.24%		

**Table 4 sensors-22-00989-t004:** Statistics of the lodged maize questionnaire answered by farmers.

Lodging Proportion (%)	Number of Questionnaires in Zhaodong City	Number of Questionnaires in Ningjiang District
81–100	5	9
61–80	4	2
41–60	7	5
21–40	0	1
0–20	2	0
Sum	18	17

**Table 5 sensors-22-00989-t005:** The change in the normalized difference vegetation index and spectral sum index between the lodged and non-lodged maize.

Index	Zhaodong City				Ningjiang District			
	Non-Lodged	Lodged	Increment	Amplification	Non-Lodged	Lodged	Increment	Amplification
Spectral sum	0.5714	0.7256	0.1542	26.99%	0.5652	0.7197	0.1545	27.34%
NDVI	0.8699	0.7886	−0.0813	−9.35%	0.7528	0.7426	−0.0102	−1.35%

## Data Availability

Gaofen1 satellite data can be obtained at http://36.112.130.153:7777/DSSPlatform/productSearch.html.

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
