# Peer review of "A Simple and Robust Spectral Index for Identifying Lodged Maize Using Gaofen1 Satellite Data"

_sensors, 2022, doi:10.3390/s22030989_

Round 1

Reviewer 1 Report

In this manuscript, authors present their work on identifying lodging or non-lodging maize area from satellite remote sensing data, and here they used a high resolution with wide-swath sensor from GF1 satellite. In their work, authors found that compared non-lodging area, the lodging maize showed higher reflectance in all blue, green, red and near-inferred bands. So they calculated a summed spectral index (SSI) to differentiate lodging area from all the images.

In general, they present their work in a clear work-flow, and it is easy to read. However, from a criterion of scientific paper, this manuscript can’t meet the requirement for publishing in this journal. There are three major problems of this manuscript.

Firstly, they reported their finding on SSI, however, there was little analysis why the SSI can be succeeded in identifying lodging area. Though they noted that higher SSI was caused by higher reflectance, it was too general and was not the basis reason. From my opinion, it might be explained from two aspects. The first one is that the canopy structure was changed by typhoon, specifically, normal canopy is featured by standing upward stem and leaves, and lodging canopy becomes as flatted and stacked leaves or stems. Flatted canopy enlarges the reflect surface and decreases the roughness of the canopy surface. The second reason is that the lodging canopy layer decreases the contribution of soil and shaded leaves which have lower reflectance, as a result the observed canopy reflectance was increased compared with non-lodging canopy.

Secondly, the structure of the manuscript should be improved. For example, in Section 3.2, when they introduced the data set, they also present some of the result in Fig.5 and Fig.6. They might want to show the reflectance was higher when maize was destructed and then thought that the proposal of SSI sounds logically. However, it will be better to theorily analysis the feature of reflectance and proposed their method according to the basis of theory. And move the two figures into the result section.

Thirdly, the English language should be improved. Although I can understand what they are talking about when reading this manuscript, many words are not properly used as literary style. For example, in abstract section, Line 12: ‘... is essential’, Line 16: ‘... to be perspective’, ‘ few studies have contributed ...’, etc. According to the language issue, It is suggested be polished by language editor before it is resubmitted.

Reviewer 2 Report

The present manuscript deals with the development of a spectral index for remotely identify lodged maize fields using satellite imagery. The present topic is of great interest to spatially assess crop yield loss due to increasing storm intensities by using open imagery data. At the same time the manuscript lacks of scientific sound, the quality of presentation is really poor with a confusing chapter structure that mix methods and results preventing to an accurate reading and understanding of the study. Superficial and often inappropriate English sentences require an extensive grammar and proof correction.

In the present form, it is really hard to follow the logic sense of the study. Methods, ground truth measurements and procedures must be clearly presented. I suggest a complete reform of the manuscript to result more focused and accurate.

Some minor comments below, even if I stopped since it would have require to correct every single sentence.

For the above reported criticism I recommend a complete rewrite of the manuscript that in the present form cannot be considered for publication in Sensors.

Abstract: The concept written at lines 26-26 have been already reported above

lines 61-62 must be rewritten

lines 68-69: this sentence is not useful and could be removed

line 70 "variation"

lines 74-87: English proof.

Lines 88-93. This paragraph must be removed and replaced by a clear statement of the aims of the study

Author Response

Thank you for your insightful comments and suggestions. Please see the attachment for our reply.

Round 2

Reviewer 1 Report

Please found in the attached file. The comments are indicated using Reviewing Tools.

Author Response

Thanks a lot. Please see the attachment.

Reviewer 2 Report

Thanks to authors' work the manuscript now results improved, focused and more concise. Before considering it as suitable for publication in Sensors, I ask the authors the following modifications:

  • the most important, in my opinion, is that the Discussion section (i.e. Section 5) still reports results, as Table 5 and Figure 9. Please move these in the appropriate Results section 4 and use the Discussion Section to properly discuss the results obtained. This, in the present state, can be improved.
  • Figure 5. I would suggest to include the meaning of the threshold dotted blu line the the caption.
  • Figures 6 and 7. I would use only Figure 6 as Figure 7 basically reports the same information. Perhaps, I suggest in Figure 6 to move NIR reflectance data for lodged and non-lodged fields in the second right Y axis thus enabling by reducing primary Y axis to better observe bloxplot and even the average symbol inside. I would include a brief description of the boxplot limits and symbols in a legend.
  • I suggest to increase Figure 8a,b legend fonts.

In the present form the manuscript requires minor modifications to be acceptable in Sensors.

Author Response

Thank you very much. Please see the attachment.

This manuscript is a resubmission of an earlier submission. The following is a list of the peer review reports and author responses from that submission.